# Reversing Type 2 Diabetes: A Narrative Review of the Evidence

**DOI:** 10.3390/nu11040766

**Published:** 2019-04-01

**Authors:** Sarah J Hallberg, Victoria M Gershuni, Tamara L Hazbun, Shaminie J Athinarayanan

**Affiliations:** 1Virta Health, 535 Mission Street, San Francisco, CA 94105, USA; shaminie@virtahealth.com; 2Indiana University Health Arnett, Lafayette, IN 47904, USA; thazbun@iuhealth.org; 3Indiana University School of Medicine, Indianapolis, IN 46202, USA; 4Department of Surgery, Perelman School of Medicine University of Pennsylvania, Philadelphia, PA 19104, USA; victoriagershunimd@gmail.com

**Keywords:** diabetes, diabetes reversal, bariatric surgery, very-low-calorie, low-carbohydrate

## Abstract

Background: Type 2 diabetes (T2D) has long been identified as an incurable chronic disease based on traditional means of treatment. Research now exists that suggests reversal is possible through other means that have only recently been embraced in the guidelines. This narrative review examines the evidence for T2D reversal using each of the three methods, including advantages and limitations for each. Methods: A literature search was performed, and a total of 99 original articles containing information pertaining to diabetes reversal or remission were included. Results: Evidence exists that T2D reversal is achievable using bariatric surgery, low-calorie diets (LCD), or carbohydrate restriction (LC). Bariatric surgery has been recommended for the treatment of T2D since 2016 by an international diabetes consensus group. Both the American Diabetes Association (ADA) and the European Association for the Study of Diabetes (EASD) now recommend a LC eating pattern and support the short-term use of LCD for weight loss. However, only T2D treatment, not reversal, is discussed in their guidelines. Conclusion: Given the state of evidence for T2D reversal, healthcare providers need to be educated on reversal options so they can actively engage in counseling patients who may desire this approach to their disease.

## 1. Introduction

According to 2017 International Diabetes Federation (IDF) statistics, there are approximately 425 million people with diabetes worldwide [1]. In the United States, there are an estimated 30.3 million adults living with diabetes, and its prevalence has been rising rapidly, with at least 1.5 million new diabetes cases diagnosed each year [2]. Diabetes is a major public health epidemic despite recent advances in both pharmaceutical and technologic treatment options. 

Type 2 diabetes (T2D) has long been identified as an incurable chronic disease. The best outcome that has been expected is amelioration of diabetes symptoms or slowing its inevitable progression. Approximately 50% of T2D patients will need insulin therapy within ten years of diagnosis [3] Although in the past diabetes has been called chronic and irreversible, the paradigm is changing [4,5]. 

The recent 2016 World Health Organization (WHO) global report on diabetes added a section on diabetes reversal and acknowledged that it can be achieved through weight loss and calorie restriction [4]. “Diabetes reversal” is a term that has found its way into scientific articles and the lay press alike; “remission” has also been used. While the exact criteria are still debated, most agree that a hemoglobin A1c (HbA1c) under the diabetes threshold of 6.5% for an extended period of time without the use of glycemic control medications would qualify [6]. Excluding metformin from the glycemic control medications list, as it has indications beyond diabetes, may also be a consideration [7,8]. Likewise, terms such as “partial” (HbA1c <6.5 without glycemic control medications for 1 year) or “complete” (HbA1c <5.7 without glycemic control medications for 1 year) remission have been defined by an expert panel as more evidence accumulates that points to the possibility of avoiding the presumably progressive nature of T2D [9]. It is important to note that the term “cure” has not been applied to T2D, as there does exist the potential for re-occurrence, which has been well documented in the literature.

Despite the growing evidence that reversal is possible, achieving reversal is not commonly encouraged by our healthcare system. In fact, reversal is not a goal in diabetes guidelines. Specific interventions aimed at reversal all have one thing in common: they are not first-line standard of care. This is important, because there is evidence suggesting that standard of care does not lead to diabetes reversal. This raises the question of whether standard of care is really the best practice. A large study by Kaiser Permanente found a diabetes remission rate of 0.23% with standard of care [10]. The status quo approach will not reverse the health crisis of diabetes.

A significant number of studies indicate that diabetes reversal is achievable using bariatric surgery, while other approaches, such as low-calorie diets (LCD) or carbohydrate restriction (LC), have also shown effectiveness in an increasing number of studies. This review will examine each of these approaches, identifying their beneficial effects, supporting evidence, drawbacks, and degree of sustainability. 

## 2. Materials and Methods

A literature search was performed as appropriate for narrative reviews, including electronic databases of PubMed, EMBASE, and Google Scholar from 1970 through December 2018. We reviewed English-language original and review articles found under the subject headings diabetes, bariatric surgery, metabolic surgery, very low-calorie diet, calorie restriction, low carbohydrate diet, ketogenic diet, diabetes remission, and diabetes reversal. References of the identified publications were searched for more research articles to include in this review. Selected studies were reviewed and evaluated for eligibility for inclusion in this review based on their relevance for diabetes reversal and remission. Either remission or reversal needed to be discussed in the paper or the results were consistent with these terms for inclusion. Randomized clinical trials and intervention-based studies were given emphasis for inclusion. 

A total of 99 original articles containing information pertaining to diabetes reversal or remission were included in this narrative review.

## 3. Results and Discussion

### 3.1. Bariatric Surgery

Bariatric surgery has long been recognized as a potential treatment for both morbid obesity and the metabolic processes that accompany it, specifically T2D. While the efficacy of T2D reversal depends on the choice of procedure, there is unilateral improvement in glycemia following operation [11], and bariatric surgery has been found to be superior to intensive T2D medical management. Accordingly, in 2016, the second Diabetes Surgery Summit (DSS-II) released recommendations, endorsed by 45 medical and scientific societies worldwide, to use bariatric surgery as a treatment for T2D (bariatric surgery is currently approved by the 2016 recommendations for adults with a body mass index (BMI) >40, or >35 kg/m^2^ with obesity-related comorbidities) [12]. Of interest is the consistent finding that glycemic improvements occur rapidly, often within hours to days, and precede weight loss, which likely represents the enteroendocrine responses to altered flow of intestinal contents (i.e., bile acid signaling and changes in microbiota and their metabolome) [13,14,15,16,17,18,19]. 

The most commonly performed bariatric surgeries in the United States include laparoscopic and robotic Roux-en-Y Gastric Bypass (RYGB) or Sleeve Gastrectomy (SG). While surgical treatment is based on the principles of restriction and intestinal malabsorption, evidence suggests that there are more complex mechanisms at play. Bariatric surgery has consistently been shown to dramatically and rapidly improve blood glucose [20] while allowing decreased oral hypoglycemic medications and insulin use, effectively reversing diabetes in up to 80% of patients [21] in the short term. In addition to early post-operative improvement in blood glucose and insulin sensitivity, bariatric surgery has also been shown to cause alterations in GI hormone release, including ghrelin, leptin, cholecystokinin (CCK), peptide-tyrosine-tyrosine (PYY), and glucagon-like peptide 1 (GLP-1), that may impact feeding behavior via the gut–brain axis in addition to modulating euglycemia [22]. Furthermore, microbial changes in the human gut have been linked to obesity, and surgical alterations to gastrointestinal anatomy have been associated with dramatic changes in gut microbiota populations with reversion from an “obesogenic” to a lean bacterial population [13,14,16,19,23,24].

Long-term outcomes from bariatric surgery depend on multiple factors, including type of surgery performed, patient comorbidities, patient readiness for lifelong dietary change, and ongoing surveillance. While bariatric surgery has been demonstrated to be safe and effective overall, it is important to recognize that it is not without risks. Each patient must weigh the risks and benefits associated with untreated morbid obesity versus those associated with surgery or effective dietary management and choose accordingly. Surgery of any type can be associated with complications leading to morbidity or mortality; the complication rates have been stated to be as high as 13% and 21% for SG and RYGB, respectively. The postoperative mortality rate is 0.28–0.34% for SG and 0.35–0.79% for RYGB; in comparison, an elective laparoscopic cholecystectomy is associated with overall complication rates of 9.29% and with a 30-day mortality rate of 0.15–0.6%, depending on the series [25,26]. Significant complications include anastomotic leak or hemorrhage, post-operative readmission, need for reoperation, post-operative hypoglycemia, dumping syndrome, worsening acid reflux, marginal ulceration, and micronutrient deficiencies [25,26,27,28,29].

It is important to consider that while short-duration studies have shown early resolution of comorbidities following bariatric procedures, when followed for multiple decades, there may be decreased efficacy of disease resolution and increased incidence of hospital admission long-term. Long-term reversal of T2D and true glucose homeostasis remain uncertain. Weight loss after surgery is a significant predictor of a return to euglycemia post-operatively. Multiple studies have reported initial T2D remission rates as high as 80% [30,31], however, long-term remission is less durable. The five-year follow-up outcomes of the SLEEVEPASS RCT found complete or partial remission of T2D in 37% of SG and 45% of RYGB patients, which is similar to other studies showing long-term T2D remission in up to a third of patients [32]. In the large prospective cohort study Longitudinal Assessment of Bariatric Surgery 2 (LABS-2), the investigators found that long-term diabetes remission after RYGB was higher than predicted by weight loss alone, which suggests that the surgery itself impacts metabolic factors that contribute to disease management [31]. Similarly, the STAMPEDE trial—an RCT that followed 150 patients with T2D who were randomized to intensive medical intervention (IMT) versus IMT plus RYGB versus IMT plus SG for diabetes resolution (defined as HbA1c <6.0%) and followed for five years—revealed increased rates of T2D resolution with RYGB (29%) and SG (23%) compared to IMT alone (5%) (Figure 1). The surgery cohort also demonstrated greater weight loss and improvements in triglycerides, HDL, need for insulin, and overall quality of life [33,34,35]. 

Despite the likelihood of improved glycemic control, there are significant financial costs for the patient, health system, and insurance companies associated with bariatric surgery (U.S. average of $14,389) [36]. Despite the high initial cost of surgery, Pories and colleagues found that prior to surgery, patients spend over $10,000 per year on diabetes medications; after RYGB, the annual cost falls to less than $2000, which represents an $8000 cost savings at the individual level [30]. Furthermore, economic analyses show that surgery is likely to be cost-effective, especially in patients who are obese [37,38]. In a clinical effectiveness review of the literature that included 26 trials extracted from over 5000 references, Picot et al. found that bariatric surgery was a more effective intervention for weight loss than non-surgical options; however, there was extreme heterogeneity and questionable long-term adherence to the non-surgical interventions [39]. After surgery, metabolic syndrome improved, and there were higher rates of T2D remission compared to the non-surgical groups [39]. Further, while there were improvements in comorbidities after surgery independent of bariatric procedure, there was also an increased likelihood of adverse events. While the overall event rate remained low, major adverse events included medication intolerance, need for reoperation, infection, anastomotic leakage, and venous and thromboembolic events [39].

It is imperative to consider that one of the requirements of qualifying for bariatric surgery is demonstration of at least six months of unsuccessful attempts at weight loss using traditional dietary and exercise advice according to the 2016 Recommendations [12]. There are, however, no requirements as to what weight loss strategy is employed, which may represent a time point where dietary intervention, including low-calorie, ketogenic, or carbohydrate-restricted diets, should be utilized. At least two recent clinical trials have demonstrated safety and efficacy in pre-operative very low-carbohydrate ketogenic diets before bariatric surgery for increasing weight loss and decreasing liver volume [40,41].

Furthermore, despite technically adequate surgery, an alarming number of patients may still experience weight regain and/or recurrence of comorbid obesity-associated conditions. In these patients, effective strategies for dietary intervention are even more important. Approximately 10–15% of patients fail to lose adequate weight (failure defined as <50% of excess weight) or demonstrate significant weight regain after bariatric surgery without evidence of an anatomic or technical reason [42]. Additionally, in 25–35% of patients who undergo surgery, significant weight regain (defined as >15% of initial weight loss) occurs within two to five years post-operatively [43]. These patients often require further medical management with weight loss medications, further dietary and behavioral intervention, and, for some, reoperation. Reoperation can be for either revision for further weight loss (narrowing of the gastric sleeve, conversion of VSG to RYGB, and increasing the length of the roux limb) or reversal of RYGB due to health concerns, most commonly associated with malnutrition. A small cohort of patients (4%) may experience severe weight loss with significant malnutrition leading to hospitalization in over 50%, mortality rates of 18%, and need for reversal of RYGB anatomy. While the incidence of RYGB reversal is unknown, based upon a systematic review that included 100 patients spanning 1985–2015, the rate of reversal parallels the increasing rate of bariatric surgery [44]. 

In the short term, T2D reversal rates with surgery have been reported to be as high as 80%, with an additional 15% demonstrating partial improvement in T2D despite still requiring medication [17]. Within one week after RYGB, patients experience improved fasting hepatic insulin clearance, reduced basal de novo glucose production, and increased hepatic insulin sensitivity; by three months and one year after surgery, patients have improved beta-cell sensitivity to glucose, increased GLP-1 secretion from the gut, and improved insulin sensitivity in muscle and fat cells [45]. Over time, T2D remission rates remain high but do decline; Purnell and colleagues reported three-year remission rates of 68.7% after RYGB [29]. However, Pories published results from a 14-year prospective study with mean follow-up of 7.6 years, and found 10-year remission rates remained around 83% [46]. In a 10-year follow-up study of participants from the Swedish Obese Subjects (SOS) study that prospectively followed patients who underwent bariatric surgery, the authors reported a 72% (*n* = 342) and 36% (*n* = 118) recovery rate from T2D for RYGB at two years and 10 years, respectively [47].

The long-term metabolic impact and risk reduction from surgery remain high in a substantial number of patients and this route to reversal clearly has the most robust data to support its use. As evidenced by the dramatic improvements in metabolic state that precede weight loss, bariatric surgery is far more than merely a restrictive and/or malabsorptive procedure. Large shifts in bile acid signaling in the lumen of the small intestine, gut nutrient sensing, and changes in the microbiota community appear to greatly impact overall host health. Further research is ongoing using both basic and translational science models to identify the role of these various hormones and metabolites; perhaps there will be a way to one day harness the beneficial effects of bariatric surgery without the need for anatomic rearrangement. 

### 3.2. Low-Calorie Diets (LCD)

As diabetes rates have risen to unprecedented levels [1,2], the number of studies examining diabetes reversal using non-surgical techniques has increased. A handful of studies have reported successful weight loss with decreased insulin resistance, plasma glucose, and medication use following a LCD. As early as 1976, Bistrian et al. [48] reported that a very low-calorie protein-sparing modified fast allowed for insulin elimination in all seven obese patients with T2D. The average time to insulin discontinuation was only 6.5 days, and the longest was 19 days. In a study by Bauman et al., a low-calorie diet of 900 kcal, including 115 g of protein, led to significant improvement in glycemic control that was mainly attributed to improvements in insulin sensitivity [49]. Furthermore, a study conducted in obese T2D patients found that a LCD and gastric bypass surgery were equally effective in achieving weight loss and improving glucose and HbA1c levels in the short term [50]. Weight loss, however, persisted in the diet-treated patients only for the first three months, indicating difficulty with long-term maintenance [47]. Similarly, other studies also reported similar pattern of early blood glucose normalization without medication use, but the improvements were not sustained long-term [51,52,53]. Likewise, the study by Wing et al., even though reported significant and greater improvements of HbA1c at 1 year in the intermittently delivered very low-calorie diet, the HbA1c improvement was not significantly different than what was reported in the patients receiving low-calorie diet (LCD) throughout the one year period [54]. Furthermore, the glycemic improvements observed at 1 year were not maintained through 2-years, even though the group with intermittent very low-calorie diet had less medication requirement than the group in the LCD arm at 2 years [54]. Lastly, micronutrient deficiencies with the use of calorie restricted diets has been shown and supplementation and monitoring for deficiencies is a consideration with their use [55,56]. 

While these previous studies were not assessing diabetes remission or reversal rate per se, they demonstrated the effectiveness of calorie restriction in achieving weight loss and improved glycemic control, which are the core goals of reversal. In 2003, the Look AHEAD trial randomized 5145 overweight or obese patients with T2D to an intervention group that received either an intensive lifestyle intervention (ILI) including calorie restriction and increased physical activity or to a control group that included diabetes support and education (DSE) [57]. Post hoc analysis of this study revealed that at one year, 11.5% of the participants in the ILI group achieved remission (partial or complete); however, remission rates subsequently decreased over time (9.2% at year two and 7.3% at year four). Nevertheless, the remission rates achieved through ILI were three to six times higher than those achieved in the DSE group. Lower baseline HbA1c, greater level of weight loss, shorter duration of T2D diagnosis, and lack of insulin use at baseline predicted higher remission rate in ILI participants [58]. 

Following the Look AHEAD study, other studies have evaluated a LCD for diabetes remission [59,60,61]. Most of these studies assessed remission over a short period of time in a small study sample. Bhatt et al. reported that six of the 12 individuals achieved partial remission at the end of the three-month intervention [61]. Ades et al. studied an intensive lifestyle program including calorie restriction and exercise, and reported that eight of the 10 individuals with recently diagnosed T2D achieved partial remission at six months, including one with complete remission [60]. The study ended at six months, therefore long term sustainability was not assessed. Another study assessing a one-year diabetes remission retrospectively among those undergoing 12 weeks of the intensive weight loss program “Why Wait” had a much lower remission rate of 4.5%, with 2.3% of them achieving partial remission, while another 2.3% had complete remission [59]. This study suggests that long-term maintenance of remission is a challenge. Moreover, diabetes remission was more likely reported in those who had a shorter diabetes duration, lower baseline HbA1c, and were taking fewer hypoglycemic medications [59,61].

An initial 2011 diabetes reversal study by Taylor and colleagues showed that a very low-calorie diet of 600 Kcal/day not only normalized glucose, HbA1c, and hepatic insulin sensitivity levels within a week, but also led to decreased hepatic and pancreatic triacylglycerol content and normalization of the insulin response within eight weeks [62]. At 12 weeks post-intervention, many of the improvements were maintained, but over a quarter of the patients had an early recurrence of diabetes. Further, average weight regain during the 12 weeks post-intervention was 20% [62]. As a follow-up to the 2011 study, the same group performed a larger and longer study with eight weeks of a very low-calorie meal replacement (624–700 kcal/day) followed by two weeks of solid food replacement and a weight maintenance program of up to six months [63]. In this study, those who achieved a fasting blood glucose of <7 mmol/L (<126 mg/dL) were categorized as responders, while others were categorized as non-responders. At six months, 40% of participants who initially responded to the intervention were still in T2D remission which was defined by achieving a fasting plasma glucose of <7mmol/L; the majority of those who remitted (60%) had a shorter diabetes duration (<4 years) [63].

These short-term studies were the foundation for a community-based cluster-randomized clinical trial called DiRECT (Diabetes Remission Clinical Trial). DiRECT enrolled a sample of 306 relatively healthy participants with T2D (people on insulin or with a diabetes duration longer than six years were excluded) [64] (Figure 1). They were cluster randomized to either standard diabetes care or an intervention using low-calorie meal replacement diet (825–853 kcal/day) for three to five months, followed by stepwise food re-introduction and a long-term weight maintenance program. At one-year follow-up, 46% of patients met the study criteria of diabetes remission (HbA1c <6.5% without antiglycemic medications) [64] and at two years the remission rate was 36% [65]. The DiRECT study has extended their follow-up an additional three years to assess the long-term impact on remission.

Taken together, evidence suggests that a LCD is effective in reversing diabetes in the short term up to two years, and its effectiveness was predominantly demonstrated in those with shorter duration since diabetes diagnosis. It is important to note that a substantial level of calorie restriction is needed to generate a sufficient level of weight loss for reversing diabetes. Short-term intervention with moderate energy restriction and metformin for modest weight loss was not as effective in reversing diabetes as compared to standard diabetes care [66]. Lifestyle intervention with severe energy restriction may have some deleterious effect on the body composition and physiology, which poses a concern for long-term health [67]. Furthermore, long-term achievement of diabetes remission, adherence to the diet, and weight loss maintenance after the diet remain a challenge. Studies have also suggested that physiological and metabolic adaptation of the body in response to caloric restriction may shift energy balance and hormonal regulation of weight toward weight regain after weight loss [67,68]. Thus, it is crucial that future studies are directed towards assessing the long-term sustainability of diabetes remission led by LCD and feasibility of this diet on the physiological adaptation and body composition changes. 

### 3.3. Carbohydrate-Restricted Diets (LC)

Before the discovery of insulin in 1921, low carbohydrate (LC) diets were the most frequently prescribed treatment for diabetes [69,70]. The paradigm shifted both with the development of exogenous insulin and later with the emergence of the low-fat diet paradigm. A diet low in fat, which by default is high in carbohydrate, became the standard recommendation in guidelines around the globe [71]. Rather than preventing elevations in glucose, the goal became maintenance of blood sugar control via the increased use of glycemic control medications, including insulin [72]. Over the last decade, clinical studies have begun to resurrect the pre-insulin LC dietary approach. In response to the new evidence on the efficacy of carbohydrate restriction, low-carbohydrate has recently been endorsed as an eating pattern by the ADA and the European Association for the Study of Diabetes (EASD) [5,73]. In addition, the Veterans Affairs/Department of Defense (VA/DOD) guidelines now recommend carbohydrate restriction as low as 14% of energy intake in its most recent guidelines for treatment of diabetes (VA) [74].

LC diets are based on macronutrient changes rather than a focus on calorie restriction [75]. Although the exact definition varies, a low-carbohydrate diet usually restricts total carbohydrates to less than 130 grams per day, while a very low-carbohydrate or ketogenic diet usually restricts total carbohydrates to as low as 20–30 grams per day. Protein consumption is generally unchanged from a standard ADA diet (around 20% of intake), with the remaining energy needs met by fat from either the diet or mobilized body fat stores. Carbohydrate sources are primarily non-starchy vegetables with some nuts, dairy, and limited fruit [75].

A total of 32 separate trials examining carbohydrate restriction as a treatment for T2D were found when our search was performed [76,77,78,79,80,81,82,83,84,85,86,87,88,89,90,91,92,93,94,95,96,97,98,99,100,101,102,103,104,105,106,107,108]. However, for reasons that may include varied levels of carbohydrate restriction and differing levels of support given, not all studies had results that would be consistent with diabetes reversal. A number of shorter-term trials have found a significant between-group advantage of a low-carbohydrate intervention for T2D [80,84,92,97]. Data from longer-term trials are limited, and in some follow-up studies, the between-group advantage seen initially was lost or reduced, although it often remains significantly improved from baseline. This raises the question of long-term sustainability using this approach. Due to heterogeneity in methodology and definition of carbohydrate restriction, the ability to fully examine T2D reversal based on the existing studies is limited. Based upon a recent systematic review of LC, it appears that the greatest improvements in glycemic control and greatest medication reductions have been associated with the lowest carbohydrate intake [109]. In consideration of these limitations, it appears important to assess the level of carbohydrate restriction, support or other methods given to encourage sustainability, and length of follow-up.

A study comparing an ad libitum very low-carbohydrate (<20 g total) diet to an energy-restricted low-glycemic diet in T2D found greater reduction in HbA1c, weight, and insulin levels in the low-carbohydrate arm [89]. Additionally, 95% of participants in the low-carbohydrate arm reduced or eliminated glycemic control medications, compared to 62% in the low glycemic index arm at 24 weeks. Instruction was given in a one-time session with a dietician and included take-home materials for reference. A slightly longer study (34 weeks) trial [85] found that a very low-carbohydrate ketogenic diet intervention (20–50 g net carbs per day) resulted in HbA1c below the threshold for diabetes in 55% of the patients, compared to 0% of patients in the low-fat arm. The education sessions were all online and included behavior modification strategies and mindful eating which was aimed to address binge eating. New lessons were emailed to the patients weekly for the first 16 weeks and then every two weeks for the remainder of the study. 

A small (34 participants) one-year study of an ad libitum, very low-carbohydrate diet compared to a calorie-restricted moderate carbohydrate diet found a significant reduction in HbA1c between groups favoring the low-carbohydrate arm [86]. At one year, 78% of participants who began the trial with a HbA1c above 6.5% no longer met the cutoff for the diagnosis of diabetes, no longer required any non-metformin medication, and significantly reduced or eliminated metformin. Total kilocalorie intake was not significantly different between the two groups, even with moderate carbohydrate restriction. Despite equal energy intake, the low carbohydrate group lost significantly more weight and had improved glycemic control, which indicates a potential mechanistic role for carbohydrate restriction itself. The support given was 19 classes over the 12-month period, tapering in frequency over time. 

Another one-year trial [76] found significant HbA1c reduction in the subset of patients with diabetes (*n* = 54) assigned to an ad libitum low-carbohydrate diet (<30 total grams per day), compared to an energy-restricted low-fat diet. These results remained significant after adjusting the model for weight loss, indicating an effect of the carbohydrate reduction itself. The support given was four weekly sessions during the first month, followed by monthly sessions for the remaining 11 months.

A metabolic ward study on 10 patients with T2D [96] found that 24-h glucose curves normalized within two weeks on a very low-carbohydrate diet (<21 g total per day). This was in addition to medication reduction and elimination including insulin and sulfonylureas After accounting for body water changes, the average weight loss during the two-week period was 1.65 kg (average of <2% total body weight which is similar to the results of bariatric surgery, where normoglycemia is seen prior to significant weight loss. Interestingly, despite the diet being ad libitum other than the carbohydrate limit, the average energy intake decreased by 1000 kcal per day. Assuming no further change in glycemic control, HbA1c would be 5.6% after eight weeks, which would represent a reduction of 23% from baseline. The fact that HbA1c reductions were greater than in other, longer-term outpatient studies may indicate that support of dietary changes is the key to longer-term success.

In our published trial providing significant support through the use of a continuous care intervention (CCI), we examined using a low-carbohydrate diet aimed at inducing nutritional ketosis in patients with T2D (*n* = 262), compared with usual care T2D patients (*n* = 87) [98] (Figure 1). At one year, the HbA1c decreased by 1.3% in the CCI, with 60% of completers achieving a HbA1c below 6.5% without hypoglycemic medication (not including metformin). Overall, medications were significantly reduced, including complete elimination of sulfonylureas and reduction or elimination of insulin therapy in 94% of users. Most cardiovascular risk factors showed significant improvement [110]. The one-year retention rate was 83%, which indicates that a non-calorie-restricted, low-carbohydrate intervention can be sustained. Improvements were not observed in the usual care patients. The newly released two-year results of this trial [106] show sustained improvements in normoglycemia, with 54% of completers maintaining HbA1c below 6.5% without medication or only on metformin. The retention rate at two years was 74%, further supporting the sustainability of this dietary intervention for diabetes reversal. Weight loss of 10% was seen at 2-years despite no intentional caloric restriction instruction. Additionally, this trial involved participants with a much longer duration of diabetes (8.4 years on average) than other nutrition trial interventions [58,64,65] and did not exclude anyone taking exogenous insulin. As duration of T2D and insulin use have both been identified to be negative factors in predicting remission after bariatric surgery [111,112], the 2-year results of this trial may be even more significant. 

It is interesting to note that most studies utilize ad libitum intake in the carbohydrate-restricted arm. Despite this, in studies that have tracked energy intake, spontaneous calorie restriction has occurred [113,114]. In many trials where energy intake has been prescribed or weight loss has been equal, an advantage has been seen in glycemic control, weight, or both in the low-carbohydrate arm [86,91,107]. A better understanding of the role that caloric intake, whether prescribed or spontaneous, plays in the overall success is important. In cases of spontaneous energy intake reduction, elucidating the specific mechanism behind this reduction would help in the overall personalization of this approach.

Multiple studies have evaluated side effects or potential complications of carbohydrate restriction. The diet has been found to be safe and well tolerated although long term hard outcome data is lacking and should be a focus of future research. A transient rise in uric acid early in very low-carbohydrate restriction without an associated increase in gout or kidney stones has been documented [84,98,100]. Blood urea nitrogen (BUN) has been found to increase and decrease in different studies without an associated change in kidney function [87,98,100,115,116]. Recently, bone mineral density has been found to be unchanged despite significant weight loss after two years of a ketogenic diet intervention in patients with T2D [108]. While most studies show an improvement or no change in LDL-C levels in patients with T2D on a low-carbohydrate diet, there have been two studies that have found an increase in LDL-C in participants with T2D [99,111]. In one of the studies that found an increase in calculated LDL-C, a non-significant reduction in measured ApoB lipoproteins and unchanged non-HDL cholesterol were seen. Monitoring LDL-C or a measured value of potentially atherogenic lipoproteins such as ApoB should be considered. Lastly, micronutrient deficiency has been seen with a carbohydrate restricted diet, supplementation and monitoring should be given consideration with this intervention [56].

Although the use of very low-carbohydrate diets for diabetes reversal shows promising results, the lack of longer-term follow-up studies remains a limitation. Follow up is limited to two years, and therefore longer-term studies are needed to determine the sustainability of the metabolic improvements. Determining the appropriate method of support may be a key to the overall success with disease reversal. 

## 4. Summary

Additional evidence has become available in recent years suggesting that diabetes reversal is a possible alternative to consider in place of traditional diabetes treatment and management. In this paper, we provide a review of three methods that have been shown to successfully reverse type 2 diabetes. The current body of evidence suggests that bariatric surgery is the most effective method for overall efficacy and prolonged remission, even though concerns associated with surgical complications, treatment cost and complete lifestyle modification after surgery remain challenges for wide adoption of this approach. While both the LCD and LC dietary approaches are convincing for reversing diabetes in the short term (up to two years), long term maintenance of diabetes remission is still unproven. There are limited available data supporting long term maintenance of weight loss and its associated glycemic improvements in response to LCD; similarly, long-term adherence to a low carbohydrate diet will likely remain an obstacle without the development of proper patient education and optimal support for long-term behavioral change. Moreover, research in understanding the mechanism of diabetes reversibility in all three approaches and its overlapping mechanistic pathways are lacking; this is an area for future research emphasis.

There are similar identified negative predictors of remission for all three approaches. These factors include longer diabetes duration and increased severity, lower BMI, advanced age, poor glycemic control, and low C-peptide levels (indicating decreased endogenous insulin production) [117]. Further exploration into the heterogeneity of these factors will help personalize the approach, determine realistic goals for each patient, and should be considered during treatment discussions. Ongoing research into algorithm development will be helpful in this regard. 

## 5. Conclusions

Overall, as a society we can no longer afford or tolerate the continued rising rates of diabetes. Despite many barriers within the healthcare system as a whole, providers are responsible on a daily basis for the lives of patients caught up in this unprecedented epidemic. The current standard of care may be suitable for some, but others would surely choose reversal if they understood there was a choice. The choice can only be offered if providers are not only aware that *reversal is possible* but have the education needed to review these options in a patient-centric discussion.

## Figures and Tables

**Figure 1 nutrients-11-00766-f001:**
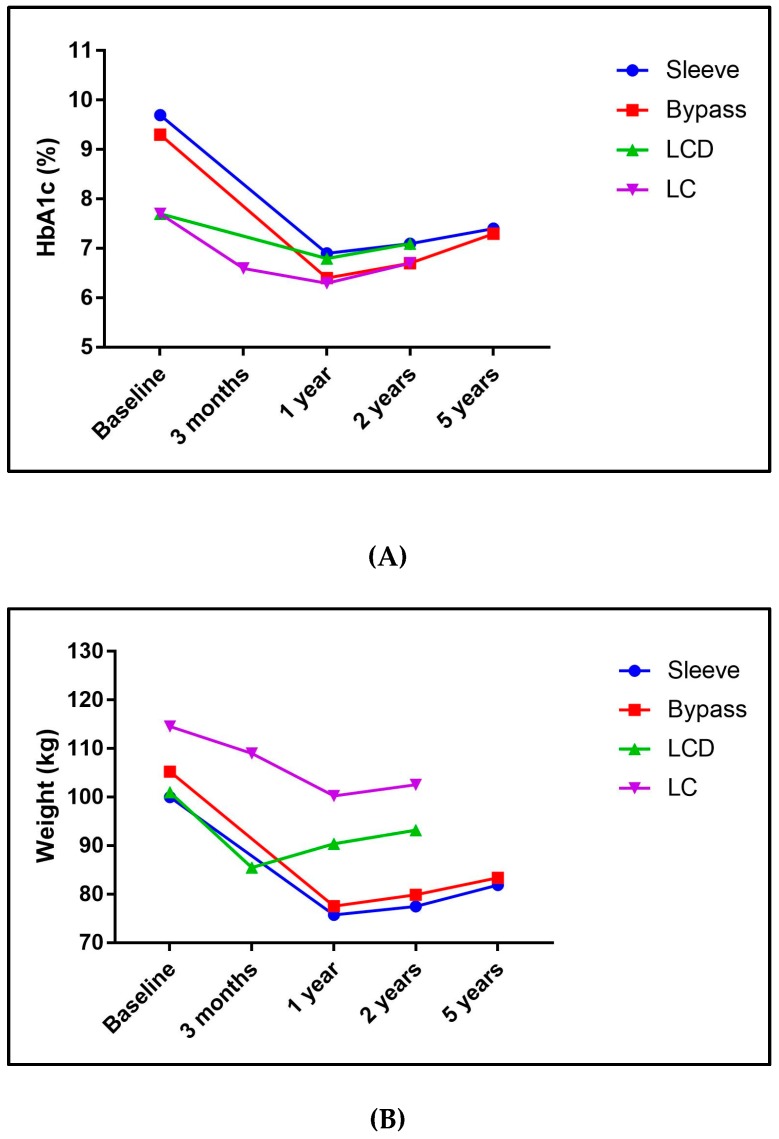
(**A**) Mean changes of hemoglobin A1c (HbA1c) from baseline to last published date for each study retrieved to represent the three methods of reversal; (**B**) mean changes of weight from baseline to last published date for each studies retrieved to represent the three methods of reversal. Note: We chose these three studies to represent the three methods of reversal based on publication date and relevance to diabetes reversal. Note that baseline characteristics differ. Surgery trial examined by sleeve gastrectomy and Roux-en-Y gastric bypass separately and were represented as sleeve and bypass in the graph. Surgery: STAMPEDE [34,35]. Low-calorie diets (LCD): DIRECT [65,66]; carbohydrate restriction (LC): IUH [99,107].

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
