# Peer review of "Reversing Type 2 Diabetes: A Narrative Review of the Evidence"

_nutrients, 2019, doi:10.3390/nu11040766_

Round 1
Reviewer 1 Report
The manuscript reviews evidence in the scientific literature showing that bariatric surgery, very low calorie diets or carbohydrate restriction can reverse type 2 diabetes, not only control or ameliorate. The manuscript challenges our current standard of care for type 2 diabetes, suggesting patients should at least have a choice knowing that diabetes may be reversal according to treatment choice. The writing is excellent, very clear and easy to understand.
As micronutrient deficiency is recognized as a major drawback of gastric bypass, it would be interesting to include a discussion on whether very low calorie diets and carbohydrate-restricted diets also presents the same concern as a care strategy to patients.
Below some minor comments that may improve the manuscript further.
- “Eligibility for inclusion” criteria in the methods should be specified.
- The conclusion is strong, however does not emphasize the
- Abstract L.20 – “Conclusion: Given”
- P. 2 L. 90 – “[19]]”
- P.7 L. 316 - What is the definition of “mindful eating”? Does it consider micronutrition as well?
- P.8 L. 365 – “released twO-year”
Author Response
See attached word file.

Reviewer 2 Report
This review gives a good, comprehensible and, overall, well readable overview of three methods for diabetes remission. However, it is not well balanced and needs to be restructured.
Criticism
It is a legitimate approach to firstly present each method individually, including description of the method, underlying mechanisms, success rate, pros and cons, and, in a second step, bring all three of them together in a short summary. If this approach is chosen, it should be consequently adhered to.
This is rather not the case in this review though: sometimes, it is referred and compared to one of the other methods, not consequently though. Also, a summary of the most important facts about each method as well as a structured comparison between the three of them is lacking.
Also, the precision and detailedness (e.g. number of n) with which the listed studies are described varies a lot within this review.
- In parts, there is too much details given that are not necessarily needed
- In others, information is lacking that would have been relevant
The discussion of low carb diets is skewed and does not represent the general state of knowledge. Recent metaanalyses particularly by Korsmo-Haugen et al., (Korsmo-Haugen et al., 2019) need to be cited and discussed.
The low carb diet discussion overstates the advantages of low carb approaches and is heavily unbalanced. The abstract and the text presents claims which are incorrect:
1.) The authors claim that the WHO supports low carb diets. In the WHO report cited (ref.4, the internet address is wrong) there is only one reference to low carb vegetables within a diabetes reversal study with a low calorie diet (Box 8). The WHO does not generally recommend low carb diets, but mentions different approaches (p.51). Therefore, this is overstated and incorrectly cited and needs to be corrected in the Abstract and in the text. Otherwise please provide the recommendation.
2.) The EASD does not recommend low carb diets. Please cite the text you are referring to. The current guideline recommend 45 – 60% carbohydrate (Mann et al., 2004).
3.) L.117: The carb diet refers to ref. 117. The list of references stops at ref. 113. Please correct.
As for the conclusion, it would be of interest to have
1. A short summary of each method
2. A clear comparison between them
a. Underlining again that “ongoing surveillance” is the crucial for each method applied and remains the currently biggest challenge.
3. A short summary of identified positive and negative predictors for T2DM remission, for each method respectively
Refs:
Korsmo-Haugen, H.K., Brurberg, K.G., Mann, J., and Aas, A.M. (2019). Carbohydrate quantity in the dietary management of type 2 diabetes: A systematic review and meta-analysis. Diabetes Obes Metab 21, 15-27.
Mann, J.I., De Leeuw, I., Hermansen, K., Karamanos, B., Karlstrom, B., Katsilambros, N., Riccardi, G., Rivellese, A.A., Rizkalla, S., Slama, G., et al. (2004). Evidence-based nutritional approaches to the treatment and prevention of diabetes mellitus. Nutr Metab Cardiovasc Dis 14, 373-394.
Reviewer 3 Report
In general this is a very useful summary of the available data. A few factual errors should be corrected:
1-Abstract, lines 18-20: in June 2018 the ADA changed this to state that remission is a recognised goal of management.
2-Line 45: This was not an officially appointed ADA group. It was as ‘expert group’ which reached a consensus. The recommendations were never accepted by ADA.
3-Line 117: Technically, DiRECT used a low calorie rather than very low calorie diet.
4- Line 242-244: The purpose of the study (Ref59) is to test if type 2 diabetes is reversible. Those patients did not receive any sort of support to maintain their weight after initial weight loss.
5-Line 249: According to the paper quoted, the statement ‘40% who initially demonstrated remission were still in remission ...’ is incorrect. The paper reports complete weight stability and complete maintenance of remission of those who initially achieved fasting plasma glucose of<7.0mmol/l.
6-Line 259-60: Could now be updated as the 2 year results have been published.
7-Line 273: The statement about safety is curious. Several studies now indicate that it is safer to embark upon calorie restriction than to continue in a diabetic state.
8-Line 293: It should be reflected by ApoB, I assume the authors meant to say ApoB containing lipoproteins.
9- Definition of remission: The authors stated that Metformin should not be counted as hypoglycaemic medication when considering diabetes remission which is concerning.
10- Carbohydrate-Restricted Diets (LC): There are no hard data about the safety of this ketogenic diet. The authors require to express a balance between the pros and cons of this method.
11- Conclusion: Very general and should highlight the major conclusions from the abstract.
Author Response
Please see attached word document
